# Carbon Dioxide Adsorption over Activated Carbons Produced from Molasses Using H_2_SO_4_, H_3_PO_4_, HCl, NaOH, and KOH as Activating Agents

**DOI:** 10.3390/molecules27217467

**Published:** 2022-11-02

**Authors:** Karolina Kiełbasa, Şahin Bayar, Esin Apaydin Varol, Joanna Sreńscek-Nazzal, Monika Bosacka, Piotr Miądlicki, Jarosław Serafin, Rafał J. Wróbel, Beata Michalkiewicz

**Affiliations:** 1Faculty of Chemical Technology and Engineering, Department of Catalytic and Sorbent Materials Engineering, West Pomeranian University of Technology in Szczecin, Piastów Ave. 42, 71-065 Szczecin, Poland; 2Faculty of Engineering, Deptarment of Chemical Engineering, Eskisehir Technical University, Eskisehir 26555, Turkey; 3Faculty of Chemical Technology and Engineering, Department of Inorganic and Analytical Chemistry, West Pomeranian University of Technology in Szczecin, Piastów Ave. 42, 71-065 Szczecin, Poland; 4Department of Inorganic and Organic Chemistry, University of Barcelona, Martí i Franquès, 1-11, 08028 Barcelona, Spain

**Keywords:** CO_2_ adsorption, activated carbon, molasses

## Abstract

Cost-effective activated carbons for CO_2_ adsorption were developed from molasses using H_2_SO_4_, H_3_PO_4_, HCl, NaOH, and KOH as activating agents. At the temperature of 0 °C and a pressure of 1 bar, CO_2_ adsorption equal to 5.18 mmol/g was achieved over activated carbon obtained by KOH activation. The excellent CO_2_ adsorption of M-KOH can be attributed to its high microporosity. However, activated carbon prepared using HCl showed quite high CO_2_ adsorption while having very low microporosity. The absence of acid species on the surface promotes CO_2_ adsorption over M-HCl. The pore size ranges that are important for CO_2_ adsorption at different temperatures were estimated. The higher the adsorption temperature, the more crucial smaller pores were. For 1 bar pressure and temperatures of 0, 10, 20, and 30 °C, the most important were pores equal and below: 0.733, 0.733, 0.679, and 0.536 nm, respectively.

## 1. Introduction

The ubiquitous global climate crisis and an improved readiness of the industrial sectors to reach a global net zero by 2050 have been concentrating on reducing greenhouse gases, especially CO_2_, which represents about 77% of all greenhouse gases [1,2]. Thus, there are developing fields of research focusing on reducing CO_2_ emissions, of which Carbon Capture, Utilisation, and Storage (CCUS) deserve special attention [3,4]. Although several promising industrial processes have already been demonstrated in various stages of development, none of the technologies can provide an economically viable and complete CCUS.

Therefore, it must be admitted that it additionally drives the ever-increasing request for energy-efficient adsorbent materials and methods for mitigating the energy footprint of chemical processes. Especially, activated carbons are considered very promising materials as CO_2_ sorbents as they meet most of the requirements: chemical and thermal stability, low energy consumption for regeneration, hydrophobicity, and stability during regeneration [5]. Remarkably, if biomass or waste is a carbon precursor, such sorbents are also inexpensive, and most biomass-based activated carbons show high adsorption capacity and selectivity [6]. The use of activated carbons as sorbents of various gases, pollutants of water, or other liquids has been described widely [7]. Moreover, these activated carbons can also be used as catalysts [8,9].

The structure and porosity of activated carbon depend on the carbon source and can be controlled by the conditions of the synthesis such as temperature, activating agent (type and quantity), duration of activation or carbonization, etc. Knowledge of how different variables influence physicochemical properties is essential and allows the design of activated carbons for desired applications, including CO_2_ adsorption. The majority of the authors demonstrate only the influence of carbonization temperature on the properties of carbon materials [10,11,12]. On the other hand, the significance of the activating agent quantity is rarely described. The most common activating agents used in activated carbon production are H_3_PO_4_, H_2_SO_4_, HNO_3_, ZnCl_2_, NaOH, and KOH [13]. Other activators, such as K_2_CO_3_, CaCl_2_, H_2_O_2_, and CH_3_NO, were used quite seldom [14].

The present study demonstrates the synthesis of activated carbon for CO_2_ adsorption from molasses using five different activating agents. Three acids: H_2_SO_4_, H_3_PO_4_, HCl, and two bases: NaOH and KOH, were used as activating agents to provide more information on the role of chemicals in the activation process during activated carbon production from biomass. Beet molasses, a sugar industry waste, was applied as the carbon source. This is a very unusual carbon precursor because it is liquid. Apart from our team’s research [15], there is only one report of obtaining activated carbon from molasses [16]. Legrouri et al. [16] produced three activated carbons from molasses for methylene blue adsorption using CO_2_ and H_2_SO_4_ as activating agents. Our team applied activated carbon from molasses activated by a saturated solution of KOH for CH_4_ adsorption [15] and activated by KOH powder for CO_2_ adsorption [17,18].

The novelty of this work is the use of molasses as a carbon precursor for CO_2_ adsorption combined with H_3_PO_4_, HCl, and NaOH as activating agents. According to our knowledge, such investigations have not been described up to now. The physicochemical properties of the activated carbons were characterized by powdered X-ray diffraction (XRD), Raman spectroscopy, scanning electron microscopy (SEM), N_2_ adsorption analysis, Fourier transform infrared reflection (FTIR), and thermogravimetry. The influence of textural properties and surface chemistry on CO_2_ adsorption was discussed.

## 2. Results

The graphitic structure and purity of activated carbons were characterized using XRD. The diffraction patterns of the activated carbons are given in Figure 1a. Activated carbons produced using acids showed two broad asymmetric peaks at 2θ, about 23 and 43°, which correspond to the plane (002) and (100/101) of the graphite structure (JCPDS 41-1487) associated with the stacking height-thickness of the layer packets (Lc) and longitudinal dimension, so-called aromatic sheets (La), respectively.

Broad peaks indicated a highly disordered carbon structure and a predominantly amorphous arrangement. Activated carbons obtained using bases showed a more disordered structure. The XRD spectra of M-KOH and M-NaOH contained only one low-intense signal. It was particularly interesting that M-KOH had a peak at 2θ ≈ 43° and M-NaOH had a peak at 2θ ≈ 23°. This means that the interaction of KOH and NaOH with carbon precursors was different. During activation by NaOH, very small aromatic sheets were produced, whilst during activation by KOH, the number of the layers in the packets was supposed to be very small.

The structural parameters obtained from XRD measurements are listed in Table 1. The interlayer spacings (d_(002)_) of the produced activated carbons are higher than that of graphite which has an interlaying spacing of 0.335 nm. For activated carbons obtained using acidic agents, the interlayer spacings were similar to each other and ranged from 0.363 to 0.368 nm. The value of d_(002)_ for M-NaOH was considerably higher (0.376 nm). It was impossible to determine d_(002)_ for M-KOH due to the absence of the peak at 2θ ≈ 23°. For the same reason, the values of Lc and the number of layers in the packets (N) could not be determined.

Considering that the number of layers in the packets for the other samples ranged from two to three, it can be postulated that there are no parallel aromatic sheets in the structure of M-KOH. The M-KOH was built of individual layers arranged in a completely disordered manner. The dimensions of so-called aromatic sheets (La) were different for activated carbon obtained using different activation agents and ranged from 3.403 to 3.878 nm. The value of La for M-NaOH was not listed in Table 1 due to the absence of a peak at 2θ ≈ 43°. It can be assumed that during the interaction of molasses with NaOH, very small aromatic sheets were produced.

Raman spectroscopy is a common method to evaluate the defects and crystallographic disorders in carbon. In Appendix A, Raman spectra with a range of Raman shifts from 500 to 2500 cm^−1^ were presented. There were two broad overlapping peaks. The first one, centred near 1330 cm^−1^ is due to the disordered portion of the carbon–D band. The second one, centred near 1600 cm^−1^ is due to ordered graphitic crystallites of the carbon (sp^2^ bonding carbon atoms)–G band. The intensities of D signals were higher than the G ones. All the spectra were normalized to the D band intensity (Appendix A). The values of the G peak maxima in Appendix A were equal to the values of the intensity ratio of the G and D bands (I_G_/I_D_). The ratios of the G band to D band intensities were compiled in Table 1.

The intensity ratio of the G and D bands allows for estimating the degree of defects. The lower I_G_/I_D_ ratio means more defects. The presence of the G band in the Raman spectra indicates that some graphene sheets can be present. The lower the intensity ratio, the smaller the size of the graphitic sheet and the higher the disorder of the graphene sheets. The values of I_G_/I_D_ ratios for activated carbon samples were quite similar and ranged from 0.686 to 0.722. The smallest values were obtained for M-KOH and M-NaOH. The structure of the M-KOH was highly disordered and probably only single graphene sheets existed, arranged concerning each other in a random manner. The dimensions of graphene sheets in the M-NaOH were so small that the XRD signal at 2θ ≈ 43° was absent. The XRD results were consistent with Raman’s findings.

Based on XRD and Raman investigations, it was found that all the samples are amorphous and disordered carbon materials, consisting of small aromatic carbon sheets stacked in 2 or 3 packets or even singular sheets.

The FTIR spectra are presented in Figure 1b. The bands in the range of 500–4000 cm^−1^ indicated the presence of carboxyl groups. The peak at 1630 cm^−1^ was attributed to C=O stretching vibrations [19]. The peak at 1120 cm^−1^ is characteristic of the coupled C–O stretching frequency and OH bending modes in the carboxylic group and the C–O stretching modes of ethers [20]. Two small peaks at around 2850 and 2920 cm^−1^ indicated the presence of symmetric and asymmetric C–H stretching vibrations, respectively, typical for hydrocarbons [21]. The wide and broad band cantered at 3444 cm^−1^ was attributed to the O–H stretching mode of H in hydroxyl groups. The hydroxyl groups can indicate the presence of phenolic or ether groups or water adsorbed on the surface [22]. For activated carbon activated by HCl, no transmittance bands were observed. The HCl treatment removed all the functional groups from the carbon surface. After treatment by basics, the C–O functional groups were nearly removed. The band characteristic of C–O was considerably broadened for M-H3PO4. The broad band at around 1000–1300 cm^−1^ was also typical for the phosphorous and phosphocarbonaceous compounds. It can be ascribed to the stretching mode of the hydrogen-bonded P=O and C–O stretching mode of the P–O–C and P=OOH groups [21]. The highest intensity of the peak at 1120 cm^−1^ being characteristic for C–O was observed for M-H2SO4 because of the overlapping bands of S=O stretching vibrations of the sulphates and sulfoxides. Moreover, a small S–O stretching vibration band was also found at 622 cm^−1^ [23].

Figure 2 shows the DTA–TG curves of activated carbons from room temperature to 1000 °C. The DTA curves of all samples show an effect with an onset of approx. 55 °C. This effect is related to the first stage of weight loss due to the removal of moisture adsorbed on the surface of activated carbons, which is also visible in the TG curves of all five samples [24]. No other thermal effects were recorded on the DTA curves of activated carbons M-HCl, M-KOH, and M-NaOH. The TG curves of the samples were quite similar. Their weight had been falling with a slow gradient with the increase in the temperature during the range of 30–900 °C, and the total weight loss ranged from 17.2 to 23.5 wt.%, depending on the sample. The weight loss in this range may be related to the decomposition of residual organic materials [25,26] due to the volatilization reactions of non-carbon functional groups. In the temperature range from 900 to 1000 °C, mass rapidly decreases by approx. 10 wt.%. The weight loss stage starts at 900 °C and can be attributed to the initiation of a progressive decomposition of the carbon [27].

On the DTA curve of the M-H2SO4 sample, apart from the effect, which is associated with the evaporation of adsorbed water, a second small endothermic effect with the beginning of approx. 250 °C was recorded. The second degree of weight loss is associated with this effect. At a temperature of 400 °C, the total weight loss is 20.5% by weight. This weight loss may be related to the removal of absorbed sulphur oxides associated with the use of concentrated sulphuric acid [23]. The presence of sulphur groups was shown in the FTIR spectrum (Figure 1b).

On the DTA curve of sample M-H3PO4, two endothermic effects were also recorded. The first effect at the onset of around 55 °C, as mentioned above, is attributed to moisture. The weight loss related to this effect recorded on the TG curve amounts to 16.3 wt.%. In the temperature range from 155 to 800 °C, there is a slight weight change. The second, slight endothermic effect, beginning at 900 °C, recorded on the DTA curve of sample M-H_3_PO_4_, is closely related to the second stage of weight loss recorded on the TG curve. At temperatures >800 °C, different P-containing compounds, and elemental phosphorus (P_4_) could be formed as a result of the reduction in phosphorus compounds previously bound to the carbonaceous solid residue [28,29]. Thus, when fewer phosphate-like structures are released, more phosphorus remains in the solid residue can be released at higher temperatures to produce the latter effect. The total weight loss of the sample M-H_3_PO_4_ recorded on the TG curve is 32.7 wt.%. The presence of phosphorus-containing groups was shown in the FTIR spectrum (Figure 1b).

SEM pictures of the activated carbons revealed the textures of the obtained materials (Figure 3 and Appendix A). The surface of activated carbon produced using bases looked as if they were made of corrugated petals. The surface of activated carbon produced by means of acids looked different. The picture showed a more massive, yet undulating surface.

Figure 4a shows the nitrogen adsorption isotherms at a temperature of 77 K. All the isotherms were characterized by rapid growth at low-pressure P/P_0_, which points that the activated carbons obtained from molasses are microporous materials. The hysteresis loops for most of the samples were very narrow, sometimes even invisible without proper magnification. The presence of the hysteresis loop indicates that mesopores also accompany the micropores. This phenomenon was the most established for activated carbon activated by sulphuric acid. Thus, it might be concluded that this modification agent contributed to the formation of the mesopores. The most rapid growth of micropores was recorded for material activated by potassium hydroxide.

According to the IUPAC classification, the three nitrogen adsorption isotherms presented in Figure 4a, except M-KOH and M-HCl, represent a composite of Types Ia and IV. The isotherm of activated carbon M-KOH was Type Ib. The ideal Ia and Ib isotherms are characterized by the relatively quick achievement of the adsorption limit, which results in the course of the adsorption isotherm being parallel to the P/P_0_ axis. The isotherms presented in Figure 4a were parallel to the P/P_0_ axis in most ranges of P/P_0_. Detailed studies on macropores have not been carried out due to the fact that these pores are not crucial for CO_2_ adsorption. They are mainly used to transport the adsorbent from the outer surface to micro and mesopores. The isotherms were reversible, but for some of them, hysteresis was observed. The hysteresis was caused by capillary condensation that took place in mesopores. The sparse hysteresis loops were identified as H3 type (for M-H_3_PO_4_ and M-NaOH), which are observed with aggregates of plate-like particles giving rise to slit-shaped pores, as well as H4 type (for M-H_2_SO_4 and_ M-NaOH) that can be correlated to narrow slit-like pores. The isotherm of M-HCl is placed on the *x*-axis in Figure 4a because the N_2_ adsorption was very low. The micropore size distribution calculated by the DFT method is presented in Figure 4b. For all the materials, pores of about 0.5 nm in diameter were dominant. The highest pore volume of the size of the pores about was 0.5 nm for M-KOH. The pore volume of these pores decreased in the following order: M-H_2_SO_4_, M-H_3_PO_4_, and M-NaOH. The second maxima were observed for a pore size of about 0.8 nm. For M-KOH, two clearly developed peaks were observed with maxima at 1.2 and 1.6 nm. For the other materials, the wide peak was observed in regions 1.1–1.7 nm. It is clearly seen that activated carbon produced using KOH exhibited the highest micropore volume. The second-highest micropore volume was found for H_2_SO_4_.

Table 2 shows the textural properties of activated carbons obtained from molasses. The highest surface area, total pore volume, and micropore volume were achieved for M-KOH. The highest mesopore volume was obtained for M-NaOH.

The values of textural properties were in good agreement with the conclusions drawn from Figure 4. The analysis of Table 1 and Table 2 indicated that the activators worked in different ways. Activated carbons produced using acids had a more ordered structure than those produced using basic, but a disordered structure was not a guarantee of high porosity.

The CO_2_ adsorption was investigated at temperatures of 0 °C, 10 °C, 20 °C, and 30 °C, up to a pressure of 1 bar. The adsorption results were presented in Figure 5a and S4. The highest CO_2_ adsorption was observed for activated carbons activated by KOH and H_2_SO_4_. The lowest values were obtained for M-HCl and M-NaOH. Some authors indicate that CO_2_ adsorption is correlated with textural properties, especially micropore volume [30,31,32]. Our results also confirmed their findings, except for M-HCl. A detailed analysis of the relationship between CO_2_ adsorption capacity at temperatures of 0 °C, 10 °C, 20 °C, and 30 °C and the pore volume of pores smaller than a specific pore size was also performed (Appendix A). The results of the best fit are presented in Table 3. The CO_2_ adsorption at a temperature of 0 to 10 °C is mainly caused by the micropores equal to or less than 0.733 nm in diameter. Further temperature increases resulted in a reduction in the pore diameter, which determines high adsorption. Similar results were presented by Deng et al. [10]. At a temperature of 20 °C and 30 °C, the CO_2_ adsorption was mainly caused by the micropores having a diameter of equal or less than 0.536 nm.

Very interesting results were obtained for M-HCl. This material was not porous. The micropore volume was equal to 0.001 cm^3^/g, pores equal to 0.733 nm and smaller were not detected, and the CO_2_ adsorption was nearly the same as for M-NaOH, while the micropore volume was a hundred times higher. It can be assumed that the reason is the absence of acid functional groups on the surface of the carbon materials. Shafeeyan et al. [33] showed that removing acidic groups from the carbon surface increased CO_2_ adsorption. To eliminate oxygen-containing acidic groups, heat treatment was applied by many authors. Such investigations were summarised in [33]. The possibility of producing activated carbon without oxygen-containing acidic groups using HCl has not been described up to now. The highest CO_2_ adsorption at 0 °C and 1 bar was equal to 5.18 mmol/g (M-KOH). This is quite a high result compared to the others that claimed high-performance CO_2_ adsorption, for example, 3.31 mmol/g at 0 °C and 1 bar. [34] and 3.2–5.3 mmol/g at 0 °C and 1 bar [35]. The CO_2_ adsorption decreased with the increase in temperature, meaning that the nature of CO_2_ sorption in over activated carbon produced from molasses is physical regardless of the activating agent.

Experimental data of carbon dioxide adsorption over the best sorbents, M-KOH and M-H_2_SO_4_, were modelled with Langmuir, Freundlich, Langmuir–Freundlich (Sips), Toth, Fritz–Schlunder, Radke–Prausnitz equations. The least-squares method was utilized as an error function. The best fittings of the experimental data were obtained for the Toth model (1).

Toth Equation (1):(1)$$q=\frac{q_{m}\mathrm{bp}}{{\left(1+{\left(\mathrm{bp}\right)}^{n}\right)}^{\frac{1}{n}}}\left[\mathrm{mmol}/g\right]$$
where

q_m_—the maximum adsorption capacity [mmol/g]

b—the Toth constant [bar^−1^]

n—the heterogeneity factor

The listed parameters above depend on the temperature according to the equations:(2)$$q_{m}=q_{m0}\cdot \exp\left[\chi \left(1-\frac{T}{T_{0}}\right)\right]$$
(3)$$b=b_{0}\exp\left[\frac{Q}{{\mathrm{RT}}_{0}}\left(\frac{T_{0}}{T}-1\right)\right]$$
(4)$$n=n_{0}+\alpha \left(1-\frac{T_{0}}{T}\right)$$
where q_m0_, χ, Q, b_0_, n_0_, and α are constants and *T*_0_ is the reference temperature [K], (273 K here).

The calculated values of constants q_m_, b, and n in the Toth Equation (1) for M-KOH and M-H_2_SO_4_ at different temperatures were presented in Appendix A. According to Equations (2)–(4), these parameters depend on the temperature. Considering Equations (2)–(4) the plots: ln(q_m_) versus T, ln(b) versus 1/T, and n versus 1/T were sketched for M-KOH (Appendix A) and M-H_2_SO_4_ (Appendix A). On the basis of these plots, the parameters of q_mo_, χ, Q, b_0_, n_0_, and α from Equations (2)–(4) were estimated and presented in Appendix A. Using Equations (2)–(4) and Appendix A, the CO_2_ adsorption over M-KOH and M-H_2_SO_4_ can be calculated at any temperature and pressure.

The isosteric heat of adsorption, i.e., the differential enthalpy of adsorption at constant coverage (θ) was calculated based on the Appendix A, after converting this equation to linear form Appendix A. The isosteric heat of adsorption was calculated by plotting ln(p)_θ_ vs. 1/T (Appendix A). The values of the pressures for given surface coverage were calculated using the Appendix A and the parameters are listed in Appendix A. Finally, the isosteric heat of adsorption as a function of surface coverage was obtained for M-KOH and M-H_2_SO_4_ (Figure 5b). The isosteric heat of adsorption decreased with the surface coverage of the activated carbon. The shape of the curves confirmed the physical nature of CO_2_ adsorption over M-KOH and M-H_2_SO_4_. A very important difference was also observed. The isosteric heat of adsorption is nearly the same up to a surface coverage of 0.1 for M-KOH, whereas for M-H_2_SO_4_ the isosteric heat of adsorption decreases very fast. The reason can be explained by the difference in porosity. The CO_2_ molecules at the initial stage of adsorption first penetrate the smallest micropores, resulting in a strong interaction between CO_2_ and the carbon surface, and hence a high isosteric heat at lower coverage was achieved. With an increase in CO_2_ adsorption, the wider pores became involved, and the CO_2_-adsorbent surface interactions became weaker. The molecules of CO_2_ covered not only the surface but also filled all the volumes of the pore. The higher the diameters of the pores, the weaker the isosteric heat of adsorption. The micropore volume for M-KOH was twice as high as than for M-H_2_SO_4_. The surface coverage was equal to 0.1. The inner surface of the smallest micropores was covered in the case of M-KOH, so the isosteric heat of adsorption was nearly constant. For M-H_2_SO_4_, the micropores were filled faster and mesopore filling took place. A similar purpose for the decrease in isosteric heat of adsorption was postulated by Abdulsalam et al. [36]. In Figure 5b, the isosteric heat of adsorption ranged from 18 to 17 kJ/mol for M-KOH and from 19 to 14 kJ/mol for M-H_2_SO_4_. Such low values of the isosteric heat of adsorption indicate that the desorption is very easy, and sorbents can be used repeatedly. The values of the isosteric heat of adsorption were lower than usually presented (22–31 kJ/mol [37], about 38.9 kJ/mol [38], and 28–18 kJ/mol [39]). The meaning of the Q parameter in the Equation (3) is the value of the isosteric heat of adsorption when the degree of coverage is approaching 0. The values of Q are presented in Appendix A (19 kJ/mol for M-KOH and 25 kJ/mol for M-H_2_SO_4_) and were consistent with the values of isosteric heat of adsorption presented in Figure 5b.

## 3. Materials and Methods

### 3.1. Materials

Molasses was purchased from the sugar factory in Kluczewo, Poland (National Food Inc., Trenton, NJ, USA).

Chemical activation of beet molasses was carried out with the use of five activating agents: HCl (Chempur 35–38%), H_2_SO_4_ (Stanlab 96%), H_3_PO_4_ (Stanlab 85%), KOH (Chempur), and NaOH (Stanlab). All reagents were used without any further purification. Carbon dioxide (99.99%) and nitrogen (99.99%) were purchased from Messer Polska.

### 3.2. Methods

Liquid molasses was weighed and then an activating agent was added in such an amount that the mass ratio of molasses to activator was 1:1. Then, the material was vigorously mixed until the raw material was clearly saturated with a saturated aqueous solution of the activating agent and left at ambient temperature for 3 h. After this time, the impregnated material was placed in a laboratory dryer (12 h, 105 °C). The carbonaceous precursor impregnated in this way was carbonized. A physical activation process was conducted in a tubular reactor kept for 1 h in an electrical furnace at a temperature of 800 °C, and the temperature was increased to 10 °C per minute to a chosen value. The process was carried out in a nitrogen atmosphere (a flow rate equal to 14.4 dm^3^/h). The activation process parameters such as time, N_2_ flow rate, and heating rate of the furnace in all the experiments were identical. Next, the derived activated carbon containing the decomposition products of activating agents was rinsed with deionized water to attain a neutral reaction. When the sample was evaporated, the activated carbon was flooded with a 1 mol/dm^3^ HCl solution and left behind for 20 h. In the following stage, carbons were rinsed with deionized water until complete removal of chloride ions. Then the samples were dried at a temperature of 105 °C for 20 h. The activated carbons were denoted as M-X, where X represents the activators: KOH, NaOH, HCl, H_2_SO_4_, and H_3_PO_4_.

Analysis of the phase composition of carbons was performed by means of a PANalytical Empyrean X-ray diffractometer (XRD) equipped with a Cu_Kα_ lamp. The obtained diffraction patterns were analysed by a comparison of the location and intensity of reflexes on the obtained diffraction patterns with the standard diffraction patterns contained in the ICDD PDF4+2015 database based on X’Pert HighScore computer software.

Raman spectroscopy was utilized to determine the structure of the carbon framework of prepared carbon materials. The analyses were performed using an apparatus equipped with a CCD detector (Renishaw InVia). The samples were induced by a laser with a wavelength of 785 nm. The spectrum was obtained in a range of Raman scattering from 800 cm^−1^ to 2000 cm^−1^. After normalization of the G peak maximum to 1, the intensity and location of the G and D peaks were assigned, and the ratio of these intensities was calculated. The ratio of G and D band intensities is generally recognized in the literature as the method for the determination of the order of the graphene layers and graphitic structure in carbon materials.

FTIR spectra of the samples were obtained by the Nicolet 6700 FT-IR spectrometer for the identification of functional groups between 500 and 4000 cm^−1^ using transmission mode. The samples were prepared using the KBr pressed-disk technique, with 1% inclusion of the material to be analysed.

To test the thermal properties of the obtained activated carbons, the samples were examined using a thermogravimetric analyser—SDT 650 DISCOVERY series (TA Instruments, New Castle, DE, USA). The tests were carried out in the temperature range from 20–25 °C to 1000 °C. The tested samples were placed in alumina crucibles, and their weight was adjusted to about 20 mg. All measurements were carried out under argon.

The morphology of the obtained activated carbons was examined using the Hitachi SU 8200 scanning electron microscope with field emission (FE-SEM).

To determine the textural properties of modified activated carbons, the low-temperature adsorption isotherms of N2 (−196 °C) were determined for the above-mentioned carbon samples by means of Sorption Surface Area and Pore Size Analyzer (ASAP 2460, Micrometrics, Novcross, GA, USA). The control and data acquisition were enabled by the ASAP software. To remove the pollutants before the adsorption measurements, the carbon samples were calcined at a temperature of 250 °C for 12 h with a heating rate of 1 °C/min under the conditions of reduced pressure.

From N_2_ sorption isotherms, the following parameters characterizing the porous structure have been determined: surface area SBET calculated from the BET equation, total pore volume V_p_ determined based on the maximum adsorption of nitrogen for a value of P/P_0_ about 0.99, micropore volume V_mi_ determined by the DFT method (Density Functional Theory), mesopores volume V_ms_ determined by the BJH method (Barrett—Joyner—Halenda).

The studies of CO_2_ adsorption for the activated carbon were carried out at the temperatures of 0 °C, 10 °C, 20 °C, and 30 °C under the pressure of 1 bar utilizing the ASAP apparatus. In order to control the measurement temperature, the sample was located in a water bath equipped with a Peltier-cooled solid-state detector. Prior to the CO_2_ adsorption measurements, the tested materials were outgassed at a temperature of 250 °C for 12 h.

## 4. Conclusions

CO_2_ adsorption at temperatures of 0 °C, 10 °C, 20 °C, and 30 °C was investigated with over activated carbons produced from molasses using H_2_SO_4_, H_3_PO_4_, HCl, NaOH, and KOH as the activating agents. All five activated carbons were obtained under the same conditions, only the activating agent was changed. The highest CO_2_ adsorption at the temperature of 0 °C and pressure of 1 bar was obtained over M-KOH (5.18 mmol/g) and M-H_2_SO_4_ (4.44 mmol/g) indicating that these materials are promising sorbents. The achieved high adsorption is even more promising because, by using molasses and KOH or H_2_SO_4_, much greater adsorption can be achieved as a result of changes in other production parameters (temperature, activating agent quantity, time of activation or carbonization, etc.).

It was stated that high microporosity promotes high CO_2_ adsorption but probably the absence of acid groups on the surface is also of great importance. Interestingly, nonporous carbon material (M-HCl) showed quite good CO_2_ adsorption results. The role and importance of HCl as a carbon-activating agent have to be developed.

The pore size ranges important for CO_2_ adsorption at different temperatures were estimated. The higher the temperature, the smaller the pores were crucial. For 1 bar pressure and temperatures of 0 °C, 10 °C, 20 °C, and 30 °C the most important were pores equal and below: 0.733, 0.733, 0.536, 0.536 nm, respectively.

The experimental data were validated with Langmuir, Freundlich, Langmuir–Freundlich (Sips), Toth, Fritz–Schlunder, and Radke–Prausnitz isotherm models. The Toth equation was found to be the best fitting and gave the lowest least-squares error.

Comparing the curves of functions E_iso_ = f(θ) for M-KOH and M-H_2_SO_4_ resulted in the isosteric heat of adsorption strongly depending on the porosity. For the activated carbons with the highest micropore volume, the decrease in the isosteric heat of adsorption was slow along with the increase in surface coverage.

Based on the isosteric heat of adsorption data and the decrement of the CO_2_ adsorption with temperature, it could be stated that CO_2_ sorption in over activated carbon produced from molasses was physical.

The mathematical description of CO_2_ adsorption characteristics (adsorption isotherms and isosteric heat of adsorption) is very important for designing an effective CO_2_ adsorption system.

## Figures and Tables

**Figure 1 molecules-27-07467-f001:**
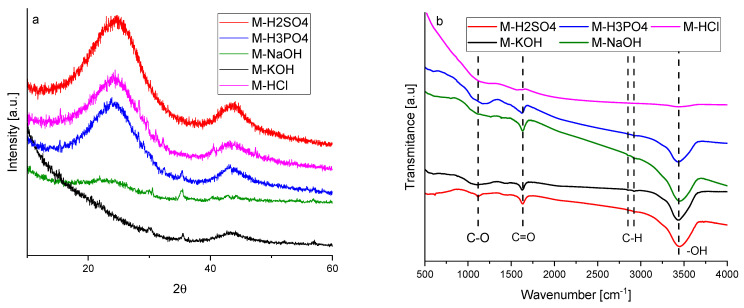
XRD diffraction patterns (**a**) and FTIR spectra (**b**) of the activated carbons produced from molasses.

**Figure 2 molecules-27-07467-f002:**
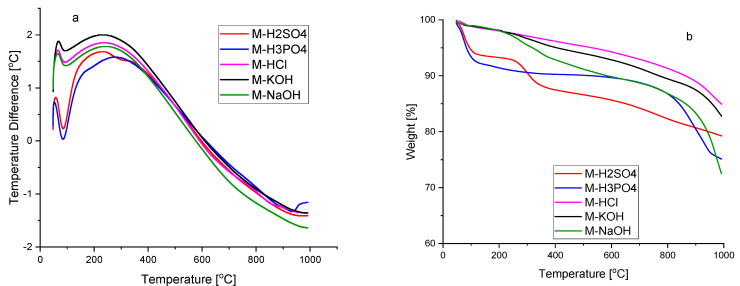
The DTA (**a**) and TG curves (**b**) of the activated carbons produced from molasses.

**Figure 3 molecules-27-07467-f003:**
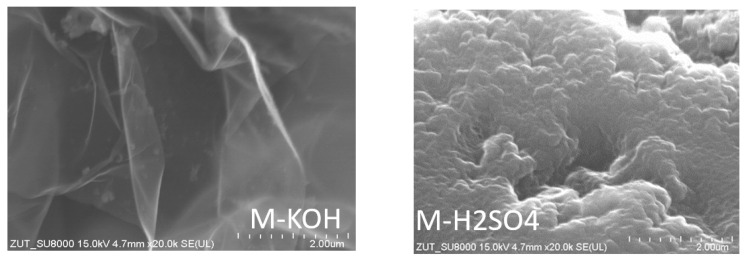
SEM pictures of the activated carbon produced from molasses.

**Figure 4 molecules-27-07467-f004:**
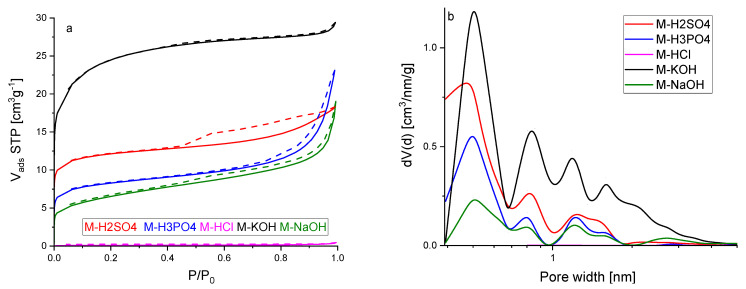
Adsorption–desorption isotherms of nitrogen (**a**) and micropore pore size distribution calculated by the DFT method (**b**) for activated carbons from molasses.

**Figure 5 molecules-27-07467-f005:**
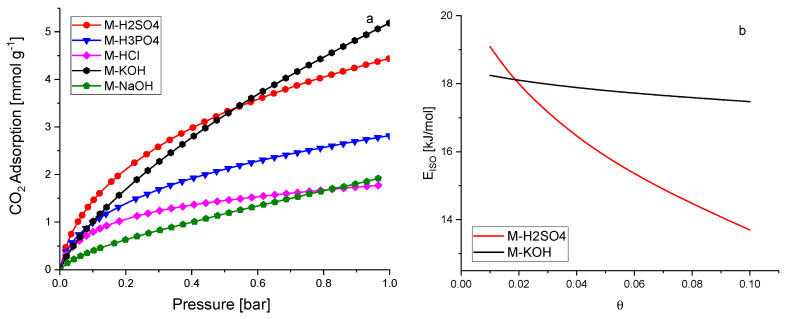
CO_2_ adsorption at a temperature of 0 °C over activated carbons from molasses (**a**), the isosteric heat of adsorption for M-KOH and M-H_2_SO_4_ (**b**).

**Table 1 molecules-27-07467-t001:** Structural parameters of activated carbons obtained from XRD, and I_G_/I_D_ ratios calculated from Raman measurements.

Activated Carbon	d_(002)_ [nm]	La [nm]	Lc [nm]	N	I_G_/I_D_
M-H_2_SO_4_	0.363	3.785	0.919	2.53	0.692
M-H_3_PO_4_	0.368	3.878	1.013	2.75	0.722
M-HCl	0.366	3.403	0.964	2.63	0.711
M-KOH		3.875			0.686
M-NaOH	0.376		1.212	3.22	0.686

**Table 2 molecules-27-07467-t002:** Textural properties of activated carbons obtained from molasses.

Activated Carbon	S_BET_ [m^2^/g]	V_p_ [cm^3^/g]	V_mi_ [cm^3^/g]	V_ms_ [m^2^/g]
M-H_2_SO_4_	1016	0.636	0.315	0.307
M-H_3_PO_4_	681	0.801	0.188	0.356
M-HCl	6	0.015	0.001	0.009
M-KOH	1970	1.020	0.635	0.362
M-NaOH	505	0.660	0.106	0.480

**Table 3 molecules-27-07467-t003:** The pore size (diameter) important for CO_2_ adsorption at different temperatures.

Temperature	Pore Size	R^2^
[°C]	[nm]	
0	0.733	0.990
10	0.733	0.994
20	0.536	0.998
30	0.536	0.993

## Data Availability

Not applicable.

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
