# Peer review of "Carbon Dioxide Adsorption over Activated Carbons Produced from Molasses Using H2SO4, H3PO4, HCl, NaOH, and KOH as Activating Agents"

_molecules, 2022, doi:10.3390/molecules27217467_

Round 1
Reviewer 1 Report
a very innovative use of waste from white sugar production to absorb carbon dioxide. It is amazing that, in principle, food production waste is useful for reducing CO2 emissions.
line 47- dot not needed before quoting 6
figure 3a sorption isotherms should have a clear distinction between adsorption and desorption isotherms, it is reasonable to use the same color but the type of lines should be different, e.g. for desorption a dashed line or a dotted line. this fact is marked in the legend or in the title of the figure Figure 3. Sorptione (adsorption ___; desorption ...) isotherms of nitrogen (a) and micropore pore size distribution
calculated by the DFT method (b) for activated carbons from molasses.
Figure 3b - for the authors to consider if it would not be better visible (more often it is) the distribution of pores on the log-log scale
Table 1 - the term "pore size" it is worth adding size is determined by diameter or by radius (r).
3. Materials and Methods - this form is not available, it is necessary to clearly distinguish materials in the text and specify the methods used in the work as subsequent subsections
line 324-325- shock! I have heard such a shallow description of the FTIR technique for the first time. It is necessary to describe whether it was a transmission or reflection technique. Were the samples analyzed in the form of tablets or directly on the FTIR ATR?
line 329 - in the analytics there is no such statement as "room temperature" in summer it will be 25 degrees C and in winter 19 degrees C.
Author Response
Dear reviewer:
The authors are very grateful for the comprehensive review of our paper and valuable suggestions how to improve our work in accordance with reviewer’s as well as journal editorial requirements.
All comments and suggestions for authors have been provided in the revised manuscript as well as they have been appended below.
Please note that line numbers that we mention in this reply refer to our revised manuscript and not to the original Molecules paper. The changed and added texts in the revised manuscript were marked up by using ‘the Track Changes’ function. The reviewer’s comments are reproduced here in bold.
Also, please note that because we added the new references in our revision, the numbering of the references in the revised manuscript is different from that in the original one.
Reviewer #1:
A very innovative use of waste from white sugar production to absorb carbon dioxide. It is amazing that, in principle, food production waste is useful for reducing CO2 emissions.
Line 47- dot not needed before quoting 6
-Authors reply
The necessary corrections have been made in the revised manuscript.
Figure 3a sorption isotherms should have a clear distinction between adsorption and desorption isotherms, it is reasonable to use the same color but the type of lines should be different, e.g. for desorption a dashed line or a dotted line. this fact is marked in the legend or in the title of the figure Figure 3. Sorptione (adsorption ___; desorption ...) isotherms of nitrogen (a) and micropore pore size distribution calculated by the DFT method (b) for activated carbons from molasses.
-Authors reply
The necessary corrections have been made in the revised manuscript.
Figure 3b - for the authors to consider if it would not be better visible (more often it is) the distribution of pores on the log-log scale
-Authors reply
You are right that Figure 3b was unreadable indeed, therefore we have incorporated the following changes: we changed X-axis for log scale in Figure 3b; log-log scale (you can find it below) looks very unusual thus we decided to use in manuscript version figure with log scale on X-axis.
Figure 3b. Micropore pore size distribution calculated by the DFT method for activated carbons from molasses
Table 1 - the term "pore size" it is worth adding size is determined by diameter or by radius (r).
-Authors reply
You are right that this was insufficient indeed, thus we have changed it and supplemented the revised manuscript with this information as you recommended.
Page 8:
Table 3. The pore size (diameter) important for CO2 adsorption at different temperatures
Also, please note that the numbering of the references in the revised manuscript is different from that in the original one.
- Materials and Methods - this form is not available, it is necessary to clearly distinguish materials in the text and specify the methods used in the work as subsequent subsections
The authors would like to thank the reviewer for pointing out this important issue. As you recommended we have been reported information about materials (page 7). Materials and Methods were divided into two subsequent subsections.
Line 324-325- shock! I have heard such a shallow description of the FTIR technique for the first time. It is necessary to describe whether it was a transmission or reflection technique. Were the samples analyzed in the form of tablets or directly on the FTIR ATR?
-Authors reply
You are right that this was insufficient indeed, thus we have changed it and supplemented the revised manuscript with this information as you recommended.
We added on page 10:
“FTIR spectra of the samples were obtained by Nicolet 6700 FT-IR spectrometer for the identification of functional groups between 500 and 4000 cm-1. using transmission mode. The samples were prepared using the KBr pressed-disk technique, with 1% in-clusion of the material to be analysed.”
Line 329 - in the analytics there is no such statement as "room temperature" in summer it will be 25 degrees C and in winter 19 degrees C.
-Authors reply
You are right that this was misleading indeed, thus we have changed it and supplemented the revised manuscript with the sentence on page 11: “The tests were carried out in the temperature range from 20 – 25 oC temperature to 1000 °C”
Thank you very much for your kind consideration of this resubmitted version of our manuscript.
Sincerely yours
RafaÅ‚ J. Wróbel et al.
Reviewer 2 Report
General comments:
-The role, influence, impact of each activating agent is not discussed in detail. Explanations of these effects should be provided.
- Why these temperatures for adsorption?
- Concentration of the activation agent, duration of the treatment and nature of the post-treatment are not provided.
- The pore size analysis giving the most important pore size is not well presented.
- Maybe too much information in the supplementary materials that would be great in the manuscript for understanding (some SEM, at least the adsorption model selected,…).
Specific comments:
- Grammatical mistakes should be corrected.
- Lines 54 to 56: references should be included.
- Lines 80, 98, 113, 124, … : References to correct.
- Line 106: Table 1 parameters should be explained (what is N (a fractional number?)).
- Line 115: why these results for NaOH and others for KOH? Explanations?
- Line 253: Table 1 again?
- Line 245-246: relationship between adsorption capacity at various T and pore volume (supposed to be constant?). How was made that analysis?
- Line 248-252: Analysis based on which results? How to interpret the 0.733 and 0.536 nm findings?
- Line 264: results from the literature should be given.
- Line 304-305: Comparison of the value vs literature?
Author Response
Dear reviewer:
The authors are very grateful for the comprehensive review of our paper and valuable suggestions how to improve our work in accordance with reviewer’s as well as journal editorial requirements.
All comments and suggestions for authors have been provided in the revised manuscript as well as they have been appended below.
Please note that line numbers that we mention in this reply refer to our revised manuscript and not to the original Molecules paper. The changed and added texts in the revised manuscript were marked up by using ‘the Track Changes’ function. The reviewer’s comments are reproduced here in bold.
Also, please note that because we added the new references in our revision, the numbering of the references in the revised manuscript is different from that in the original one.
Reviewer #2:
General comments:
-The role, influence, impact of each activating agent is not discussed in detail. Explanations of these effects should be provided.
-Authors reply
The authors would like to thank the reviewer for pointing out this important issue. Indeed, the authors agree with the reviewer that the detailed discussion would be very useful for the evaluation. However, we presented here the preliminary results of the use of different activating agents. On this stage, it is impossible to discuss the role, influence, impact of each activating agents. Certainly, we will perform this investigations and the results will be described by us in the future. Meanwhile, we kindly ask for your understanding. Since all comments and suggestions of the reviewers have been addressed in the revised manuscript we hope that the reviewer and the editor will find the paper suitable for publications.
- Why these temperatures for adsorption?
-Authors reply
In our laboratory, we have such research possibilities: from 0 – 30 oC. Temperature of 0 oC and 20 oC are the most applied temperatures for CO2 adsorption. Moreover, we applied 10 oC, and 30 oC in order to make calculations of the isosteric heat of adsorption more precisely.
- Concentration of the activation agent, duration of the treatment and nature of the post-treatment are not provided.
-Authors reply
The necessary corrections have been made in the revised manuscript (page 10-11).
- The pore size analysis giving the most important pore size is not well presented.
-Authors reply
You are right that this was insufficient indeed, thus we have changed it and supplemented the revised manuscript with this information as you recommended.
We added on page 7:
“For all the materials pores about 0.5 nm diameter were dominant. The highest pore volume of the pores size about 0.5 nm was obtained for M-KOH. The pore volume of these pores decreased in the following order M-H2SO4, M-H3PO4, M-NaOH. The second maximum was observed for pore size about 0.8 nm. For M-KOH two another clearly developed peaks were observed with maxima at 1.2 and 1.6 nm. For the other materials the wide peak were observed in region 1.1 – 1.7 nm.”
- Maybe too much information in the supplementary materials that would be great in the manuscript for understanding (some SEM, at least the adsorption model selected,…).
-Authors reply
You are right that this information was unbalanced indeed, we added in the revised manuscript some SEM and the Toth model description as well.
Specific comments:
- Grammatical mistakes should be corrected; Lines 54 to 56: references should be included; Lines 80, 98, 113, 124, … : References to correct.
-Authors reply
The necessary corrections have been made in the revised manuscript.
- Line 106: Table 1 parameters should be explained (what is N (a fractional number?)).
-Authors reply
N is the number of the layers in the packets; (It was on page 3 in line 104-105);
- Line 115: why these results for NaOH and others for KOH? Explanations?
-Authors reply
The authors would like to thank the reviewer for pointing out this important issue. It is not obvious that since KOH and NaOH are strong bases they must affect molasses in a similar way. The properties depends on the source of carbon.
Have a look at Journal of Colloid and Interface Science 298 (2006) 341–347. Authors used KOH and NaOH for activation of commercial impregnation pitch, petroleum pitch and synthetic commercial mesophase pitch.
In case of commercial impregnation pitch the specific surface area of sample activated by NaOH was 20% higher. For petroleum pitch the specific surface area of sample activated by KOH was 60% higher. The molasses is completely different material. Firstly is liquid. More investigations is needed to answer why is a big difference. We present here preliminary research. The work will be continued.
- Line 253: Table 1 again?
-Authors reply
The necessary corrections have been made in the revised manuscript (it should be Table 3).
- Line 245-246: relationship between adsorption capacity at various T and pore volume (supposed to be constant?). How was made that analysis?
-Authors reply
Adsorption capacity vs pore volume supposed to be straight line. The analysis is presented in Figures S 5 – S 8.
- Line 248-252: Analysis based on which results? How to interpret the 0.733 and 0.536 nm findings?
-Authors reply
We explained it on page 8:
A detailed analysis of the relationship between CO2 adsorption capacity at temperatures of 0, 10, 20, and 30 oC and pore volume of pores smaller than a specific pore size was also performed (Figure S 5 – S 8).
The values of 0.733 and 0.536 nm are difficult to interpret but they are consistent with results of the other authors. They also couldn’t find any interpretation.
- Line 264: results from the literature should be given.
-Authors reply
The results were added.
- Line 304-305: Comparison of the value vs literature?
-Authors reply
The comparison was added (page 9). The values of the isosteric heat of adsorption were lower than usually presented (22–31 kJ/mol [https://doi.org/10.1016/j.fuel.2021.122507], about 38.9 kJ/mol [https://doi.org/10.1016/j.scitotenv.2022.157805], 28 – 18 kJ/mol [https://doi.org/10.1016/j.micromeso.2022.111801]);
Thank you very much for your kind consideration of this resubmitted version of our manuscript.
Sincerely yours
RafaÅ‚ J. Wróbel et al.
Round 2
Reviewer 2 Report
My previous comments has been answered but investigating the role, influence, impact of each activating agent is not discussed in detail. Explanations of these effects would be interesting in another publication.
Author Response
Dear reviewer,
thank you for understanding that we are not able to discuss that role, influence, impact of each activating agent now.
Minor spell check was made.
Thank you very much for your kind consideration of this resubmitted version of our manuscript.
Sincerely yours
RafaÅ‚ J. Wróbel et al.
